# Measuring S-Phase Duration from Asynchronous Cells Using Dual EdU-BrdU Pulse-Chase Labeling Flow Cytometry

**DOI:** 10.3390/genes13030408

**Published:** 2022-02-24

**Authors:** Marta Bialic, Baraah Al Ahmad Nachar, Maria Koźlak, Vincent Coulon, Etienne Schwob

**Affiliations:** 1Institut de Génétique Moléculaire de Montpellier, Univ Montpellier, CNRS, 34293 Montpellier, France; marta.bialic@igmm.cnrs.fr (M.B.); bnachar@igmm.cnrs.fr (B.A.A.N.); maria.kozlak@igmm.cnrs.fr (M.K.); etienne.schwob@igmm.cnrs.fr (E.S.); 2Institut de Médecine Régénératrice et Biothérapie, INSERM, CHU, 34295 Montpellier, France

**Keywords:** DNA replication, S phase, EdU-BrdU pulse chase, cytometry, cell lines

## Abstract

Eukaryotes duplicate their chromosomes during the cell cycle S phase using thousands of initiation sites, tunable fork speed and megabase-long spatio-temporal replication programs. The duration of S phase is fairly constant within a given cell type, but remarkably plastic during development, cell differentiation or various stresses. Characterizing the dynamics of S phase is important as replication defects are associated with genome instability, cancer and ageing. Methods to measure S-phase duration are so far indirect, and rely on mathematical modelling or require cell synchronization. We describe here a simple and robust method to measure S-phase duration in cell cultures using a dual EdU-BrdU pulse-labeling regimen with incremental thymidine chases, and quantification by flow cytometry of cells entering and exiting S phase. Importantly, the method requires neither cell synchronization nor genome engineering, thus avoiding possible artifacts. It measures the duration of unperturbed S phases, but also the effect of drugs or mutations on it. We show that this method can be used for both adherent and suspension cells, cell lines and primary cells of different types from human, mouse and *Drosophila*. Interestingly, the method revealed that several commonly-used cancer cell lines have a longer S phase compared to untransformed cells.

## 1. Introduction

Faithful genome replication is required at every cell division during cell proliferation and tissue development. Cells have evolved several mechanisms to regulate in space and time the activation of replication origins, and to adjust the speed of replication forks according to local or cellular demands. The reduplication and amplification of genome segments, for example, is prevented by a mutually-exclusive two-step mechanism of origin licensing and origin firing. These two steps are temporally separated in a low cyclin-dependent kinase (CDK) activity during late M and G_1_ phases when pre-replication complexes (pre-RCs) assemble on origins (the licensing step), and a high CDK activity S phase in which origins are activated (the firing step) and pre-RCs disassembled [1,2]. Once origins are activated, bi-directional DNA synthesis takes place at replication forks progressing at 1–2 kb/min, along with dozens of proteins that regulate synthesis rate, protect forks and restore chromatin on daughter DNA molecules [3,4,5,6]. Co-regulated origins are clustered in space as discrete foci associated with different states of chromatin. Each of these highly controlled processes can be targeted by oncogenic mutations, resulting in processes collectively referred as replication stress: re-replication, fork collapse, origin paucity, nucleotide deprivation and interference with transcription [7]. Consistently, a number of cellular models of tumor initiation or genome instability (yeast *sic1Δ* mutants, Cln2 or cyclin E-overexpressing cells, Myc-depleted cells) display an extended S phase [8,9,10,11], also suggesting that they undergo replication stress. Importantly, while the two-step mechanism described above ensures that origins are activated once and only once, it also implies that origins that fail to be licensed in G_1_ will never fire, hence lengthening S phase or leaving some regions un-replicated. Although often postulated, the very existence of a replication completion checkpoint has been questioned in yeast cells, which can divide without replicating [12,13] or at human chromosome fragile sites [14]. More specifically, it seems that un-replicated DNA fails to activate the checkpoint [8,15,16]. If verified in mammalian cells, this phenomenon would be of importance in the context of origin-poor regions, which in case of replication stress would have to be replicated by forks traveling from distant origins. Critically, Common Fragile Sites have been correlated with origin-poor late-replicating chromosomal regions [17]. The importance of replication stress is not limited to oncogenesis: cultured pluripotent cells [18] and reprogrammed iPSCs [19] undergo replication stress, which is considered as a barrier to re-programming and a source of genomic instability, both being important issues for regenerative medicine. Therefore, a thorough measure of S-phase duration (SPD) in all these models would be of interest, both to better characterize their cell cycle and to estimate their level of replication stress.

Traditional methods for measuring SPD include simple DNA content flow cytometry followed by modeling the 2N and 4N cell distribution to determine the S-phase fraction (SPF), which is then multiplied by the population doubling time [20,21]. However, population-doubling times are muddled by the fraction of non-cycling or dying cells in the culture, and only allow an inaccurate estimation of SPD. While bivariate bromo-deoxyuridine (BrdU)/DNA content analysis alleviates some uncertainties, it still measures the SPF only [22]. Alternatively, measuring cell division times by video-microscopy can exclude non-cycling cells, but this approach is tedious, subject to phototoxicity and will not remove the bias introduced by the highly variable G_1_ length in cell culture. Other methods use a brief BrdU pulse and then follow by flow cytometry the movement of the cloud of BrdU^+^ cells over time [23,24,25], but determining exactly when these cells reach G_2_ is difficult and potentially leads to an overestimation of the SPD. Finally, the direct observation of cells progressing through S phase by video-microscopy of fluorescent markers of DNA replication such as PCNA-GFP foci [26,27,28] or cell cycle reporters [29] is informative but is poorly sensitive and requires genome editing, or the use of cell-permeant replication tracer [30]. In both cases, exposure to strong light or tinkering with replication factors may alter replication dynamics and inaccurately determine the SPD of non-edited cells.

We describe here a novel method to directly measure SPD in cultured cells. The method is simple, robust, applicable to various metazoans including invertebrates, and does not require cell synchronization or genome editing. It is based on a first pulse labeling with 5-ethynyl-2’deoxyuridine (EdU) to single out cells in S phase, followed by thymidine chases of increasing length and a second pulse with 5-bromo-2’-deoxyuridine (BrdU) to capture cells that are still in S at this time. Monitoring the decreasing fraction of double-positive cells over time scores cells exiting from S, and extrapolating the time when this fraction becomes null provides the duration of S phase. It is similar in design to older methods using H^3^dT-BrdU dual labeling [31], but faster and scorable by flow cytometry. We believe that this method will be useful to any laboratory interested in cell cycle and DNA replication, in various species including invertebrates, on adherent or suspension cells, or in the context of normal, cancer or pluripotent cells. During the validation of this method, we discovered that commonly used cancer cell lines have a lengthy S phase relative to primary cells, which is counterintuitive given their faster proliferation.

## 2. Materials and Methods

Cell culture: S2R+ Drosophila cells were cultured at 28 °C in Schneider medium (Pan Biotech, Dutscher, Bernolsheim, France), with 10% FCS (Pan Biotech) and penicillin/streptomycin. Other cell lines were cultured at 37 °C, with 5% CO_2_, and for Mouse Embryo Fibroblasts, O_2_ was maintained at 2% in an MCO-5M tri-gas incubator (Sanyo, Avon, France). DMEM with Glutamax supplemented with 10% FCS (Sigma, St Quentin Fallavier, France or Pan Biotech, Dutscher, Bernolsheim, France) and antibiotics (penicillin/streptomycin, Life Technologies, Courtaboeuf, France) was used for adherent mammalian cell lines. Human primary T lymphocytes were purified from healthy donor peripheral blood, activated on TCR + CD28 coated 24-well plates, and grown in RPMI medium with 10% inactivated FCS (PAN Biotech, Dutscher, Bernolsheim, France), glutamine and carbonate, as for the Jurkat cell line.

BrdU/7-AAD cytometric analysis: After incubation with 100 µM BrdU (Sigma, St Quentin Fallavier, France) for 1 h, 10^6^ cells were trypsinized, resuspended in cold PBS and resuspended carefully in 200 µL cold PBS. For fixation, 5 mL EtOH 95% was added dropwise while vortexing at low speed. After washing with PBS, 1 mL 2N HCl with 0.5% Triton X100 was added dropwise while vortexing, cells were incubated at 25 °C with occasional shaking, then rinsed with PBS. Neutralization with 1 mL 0.1 M borate buffer and a PBS rinse were performed before adding the mouse anti-BrdU antibody (Becton Dickinson B44, diluted 1:30 in PBS 0.5% Tween20, 1% BSA). After 2 h incubation at 25 °C, cells were rinsed in PBS and incubated 1 h with Alexa 488-conjugated anti-mouse antibody (1:300, Life Technologies, Courtaboeuf, France) diluted 1:40 as above. Finally, cells were rinsed in PBS and resuspended in PBS containing 14 µg/mL 7-amino actinomycin D and 200 µg/mL RNase A (Sigma). Right before flow cytometry, cells were passed through a 70 µm nylon mesh to eliminate aggregates. Flow cytometry was performed on FACSCalibur II (Becton Dickinson, Le Pont-de-Claix, France).

S-phase duration, cell labeling and detection: Several dishes of cells (each containing 10^6^ cells) were incubated with 10 µM EdU (Carbosynth, Compton, United Kingdom) for 30 min, the medium removed and replaced with new medium containing 20 µM thymidine (dT) and 100 ng/mL nocodazole (Sigma, St Quentin Fallavier, France) for either 0, 2, 4, 6, 8, 10 or 12 h, then the medium was changed again and cells incubated 30 min in medium containing 100 µM BrdU (Sigma, St Quentin Fallavier, France). For S2R+ cells, nocodazole was replaced by 30 µM colchicine (Sigma, St Quentin Fallavier, France). After the BrdU pulse, cells were detached with trypsin/EDTA, rinsed in PBS, fixed with cold 95% EtOH added dropwise while vortexing, then treated with HCl 2N and 0.5% Triton X-100, and neutralized with 0.1 M Borate buffer pH 8.5. After rinsing twice with PBS-1% BSA, EdU was detected using click reaction for 45 min at RT with 10 µM disulfo-cyanine-azide (CyanDye, Miami, United States), in PBS-1% BSA containing 8 mM CuSO_4_ and 40 mM ascorbic acid. Cells were rinsed twice with PBS containing 0.5% Tween20 and 1% BSA, then stained with 1 µg/mL anti-BrdU antibody (MoBU-1, Thermo Fisher, Illkirch, France; other anti-BrdU antibodies used include clone Bu20a from DAKO (Les Ulis, France), B44 from Becton Dickinson (Le Pont-de-Claix, France), IIB5 from Santa Cruz (Heidelberg, Germany) as indicated) for 2 h at 25 °C, then with 2 µg/mL anti-mouse antibody coupled to Alexa Fluor488 (Life Technologies, Courtaboeuf, France) for 1 h at 25 °C. After two PBS rinses, DAPI was added to 1 µg/mL and cells were passed through a 70 µm nylon mesh before flow cytometry.

Flow cytometry for SPD measurements: A FACSCanto II (Becton Dickinson, Le Pont-de-Claix, France) with 405, 488 and 633 nm laser lines was used. When the signal-to-noise ratio was sufficient (more than 1 log_10_), EdU vs BrdU signals were plotted and the fraction of EdU-positive cells that were also BrdU-positive (EdU^+^BrdU^+^/EdU^+^) representing the percentage of cells still in S phase after the second pulse was plotted as a function of time. Linear regression of aligned values was used to extrapolate S-phase duration. Later time points (10–12 h) having EdU^+^BrdU^+^/EdU^+^ values approaching zero were excluded from regression analysis, as they may represent a small fraction of cells either progressing very slowly through S or some cells entering the subsequent S phase. Considering this non-linear part of the curve would significantly overestimate the SPD of the major population.

A step-by-step protocol is provided in Appendix B and Appendix C.

## 3. Results

### 3.1. Rationale of the Method

EdU and BrdU are two thymidine analogs that are naturally taken up by most cell types through the nucleotide salvage pathway, phosphorylated by thymidine kinase into EdUTP and BrdUTP that are then incorporated by DNA polymerases into DNA at the sites of DNA synthesis (replication, recombination, repair). BrdU and anti-BrdU antibodies have been used for decades for replication analyses, whereas EdU has been introduced more recently with the advantage of easy, efficient and bio-orthogonal detection by copper-catalyzed alkyne-azide cycloaddition (CuAAC, or Click chemistry) using small fluorescent azide dyes [32]. In theory EdU and BrdU should be detected with excellent specificity. With this in mind, we thought to first label briefly the fraction of cells in S phase with EdU, chase it with thymidine for varying times, and finally, label the cells that are still in S phase with BrdU. Without intervening thymidine chase most cells should be labeled with the two analogs, but after increased chase times the fraction of double-positive cells should gradually decrease as cells exit S phase (Figure 1a,b).

The time required for the earliest S-phase cells not to be labeled with the second analog corresponds to the duration of the S phase (Ts). Nocodazole, which prevents spindle formation, is added along with thymidine to prevent cells from entering mitosis and getting labeled with BrdU in the subsequent S phase. Provided that each analog can be detected specifically, it would be straightforward to quantitate by flow cytometry the fraction of double-positive cells, their decrease with longer chase times, as well as the flux of cells entering and exiting the S phase (Figure 1c).

### 3.2. Selection of a BrdU Antibody Not Cross-Reacting with EdU

The method relies on optimal discrimination between EdU-positive and BrdU-positive cells. Therefore, using an antibody that does not cross-react with EdU is paramount. We thus tested several commercially available anti-BrdU antibodies by staining mouse embryonic fibroblasts (MEF) previously labeled with EdU (10 µM, 30 min) and flow cytometry (Figure 2). All but one of the four BrdU antibodies we tested cross-reacted strongly with EdU. Only the MoBU-1 antibody (ThermoFisher) was specific for BrdU with little or no detection of EdU (Figure 2l). This, in agreement with earlier studies [33,34], prompted us to use the MoBU-1 antibody for further experiments.

### 3.3. Choice of Nucleoside Concentrations for Dual Pulse

Another requirement for this method is that exogenous nucleosides are incorporated at a level allowing good detection, but also that they are chased efficiently during the pulse. We performed competition experiments, which indicated that BrdU is incorporated preferentially over EdU (not shown). Therefore, we decided to use EdU for the first pulse to ensure optimal detection of the critical second pulse. The fractions of EdU^+^ and BrdU^+^ cells were detected by Cy5-azide Click chemistry and MoBU-1/AF488-coupled antibodies, respectively, and flow cytometry. A 30 min pulse with EdU (10 µM) was found sufficient for a clear distinction between EdU^+^ and EdU^−^ cells by cytometry, with a signal to noise ratio ≥ 20 (Figure 1c). After changing the culture medium, a 2-fold higher concentration of thymidine (20 µM) was found sufficient to cancel the increase in the fraction of EdU^+^ cells with time, and thus to effectively terminate EdU incorporation (not shown). Finally, a concentration of 100 µM BrdU was required in the second pulse (after another change of medium) to overcome the thymidine present in cells while giving a clear BrdU^+^ signal by FACS (Figure 1c). Since thymidine chases were performed for periods up to 8–10 h, nocodazole (100 ng/mL, 0.33 µM) was added to prevent cells from entering a new cell cycle. The efficacy of mitotic arrest was confirmed by the absence of BrdU^+^ cells after long periods (14–18 h) of chase.

By separating the first (EdU) and second (BrdU) pulses by a defined time and plotting BrdU versus EdU signal intensity, one can determine the fraction of cells that have exited S (EdU^+^ BrdU^−^) and entered S (EdU^−^ BrdU^+^) during this time (Figure 1c).

### 3.4. Linear Regression of EdU^+^ BrdU^+^ Cells over Chase Time Determines SPD

In exponentially growing cell populations, the fraction of EdU^+^ cells that are also BrdU^+^ is 100% when the two pulses are performed simultaneously, but should decrease regularly when thymidine chase times increase, upon S-phase exit of initially-labeled EdU^+^ cells (Figure 1b). This fraction of double positive cells (EdU^+^ BrdU^+^/EdU^+^) should, by definition, reach zero when the time separating the two pulses corresponds to the time required for the earliest EdU^+^ cell to complete S phase. This time defines S-phase duration (SPD, or Ts).

To put this idea to the test, we labeled an asynchronous culture of MEFs with EdU for 30 min, then introduced (or not) a thymidine chase of 2 h, 4 h, 6 h, 8 h, 10 h, or 12 h and nocodazole to prevent cells moving to the next cell cycle, before the second pulse of BrdU (30 min). Cells were treated and analyzed by FACS as indicated above. Figure 3 shows the fraction of EdU^+^ BrdU^+^ cells (top-right quadrant) decreases with increasing chase times, as expected. When the fraction of EdU^+^ BrdU^+^ over EdU^+^ cells was plotted as a function of time between the two pulses, the values between 0.5 h and 8 h fitted well a regression line (R^2^ = 0.99) whose intersection with the x-axis (8.94 h) defines the duration of S phase (Ts, Figure 3h). No more EdU^+^ cells were BrdU^+^ after this time, indicating that S phase was completed and that cells did not enter the S phase of the subsequent cell cycle.

The ability to follow the population of S-phase cells as they progress through and exit S over time allows determining the duration of S phase with good precision. Moreover, the good linear regression fitting indicates that S-phase progression was uniform in the cell population. The method is robust; the mean (±SEM) S-phase duration was found to be 8.94 h (±0.22, *n* = 5) for MEF.

### 3.5. Compatibility with Mammalian Cells from Different Origins and Drosophila Cells

After setting up the conditions and validating the method section on MEF cells, we wanted to check if it was applicable to other cell types and species. The same concentrations of EdU (10 µM), thymidine (20 µM) and BrdU (100 µM) were effective in the other cell lines tested (Appendix A). Indeed, we succeeded in measuring SPD in primary human BJ fibroblasts (8.59 h ± 0.21, *n* = 3), as well as in the NIH 3T3 mouse immortalized fibroblast cell line (10.52 h ± 0.55, *n* = 3). In addition to these adherent cells, we determined the SPD of human primary lymphocytes from peripheral blood (12.40 h ± 0.36, *n* = 2), showing that suspension cells also can be used (Figure 4a). Finally, as *Drosophila melanogaster* is a significant invertebrate model, we tested if S2R+ cells were also amenable to SPD measurements using our method. Since these cells are known to be insensitive to nocodazole [35], we replaced the latter with colchicine. Because of the short S phase duration of S2R+ cells, accurate linear regression was built by harvesting samples every hour for 6 h. The rest of the protocol was unchanged and gave an SPD value of 5.67 h (±0.31, *n* = 6), as predicted by other studies [36,37].

### 3.6. Commonly-Used Cancer Cell Lines Have a Longer S Phase

Tumor cells frequently harbor mutations that deregulate the G_1_ phase of the cell cycle and may affect the dynamics of S phase. Replication stress has also been proposed as a driving force in tumorigenesis [38]. We applied our SPD measurement method to see whether commonly-used cancer cell lines have a normal or deregulated S phase duration. Interestingly, the S phase lasted significantly longer (11–13 h) in carcinoma-derived HCT116 and HeLa cells, as well as in osteosarcoma-derived U2OS and leukemia Jurkat cells, compared to untransformed primary cells (Figure 4b, Table 1). For example, HeLa cells had a mean (±SEM) S-phase duration of 13.6 h (±1.35, *n* = 4) compared to 8.59 h (±0.22, *n* = 3) in normal BJ fibroblasts. T-cell leukemia Jurkat cells also displayed a long SPD (12.66 h ± 0.77, *n* = 2), but in this case, it was similar (12.4 h ± 0.36, *n* = 2) to human primary lymphocytes obtained from healthy donors and activated by TCR-CD8^+^ stimulation. It is possible that the long SPD in the latter cells stems from their slowed cell proliferation observed 4–6 days after stimulation, or from antigen-induced cell death (AID) caused by strong TCR activation [39,40]. Nonetheless, our finding that several cancer cell lines display a prolonged S phase is unexpected given their fast proliferation, and deserves further investigation.

### 3.7. S-Phase Extension by Replication Stressors

The method described here to measure S-phase duration being both precise and robust, we next considered using it to detect and quantify the extent of replication slowing down by drugs. To this end we exposed MEF cells to low doses (0.2 or 0.6 µM) of aphidicolin, an inhibitor of B-family DNA polymerases, which slow down the progression of replication forks without activating the DNA damage response [41].

We confirmed that aphidicolin, at these low doses, does not prevent DNA synthesis but significantly and quantitatively reduced the rate at which EdU^+^ MEF cells exited S phase, compared to untreated control cells (Figure 5). This reduction was greater with 0.6 µM than with 0.2 µM aphidicolin, as expected. Linear regression and extrapolation of these values lead to an estimated S phase of 31.11 h (±1.62, *n* = 2) at 0.2 µM and of 123.9 h (±25, *n* = 2) at 0.6 µM aphidicolin. These doses of aphidicolin reduce fork speed 80% and 90%, respectively, in JEFF lymphoblastoid cells, or 65% and 80% in MRC5 fibroblasts, without activating the Chk1- and ATM-dependent DNA damage responses [41]. Our estimation of S-phase duration in aphidicolin-treated MEF cells is grossly proportional to these reduced fork speeds, suggesting that dormant origin activation does not deter S-phase extension in such conditions, consistent with the decreased mitotic flow observed by these authors. We conclude that our SPD measurement method can be used to assess the degree of fork slowing, replication stress and how cells manage (or not) to deploy compensating strategies to limit the risk of S-phase extension and genome instability.

## 4. Discussion

We describe here a new method, based on a dual EdU-BrdU pulse protocol and flow cytometry, to measure the duration of the S phase in asynchronously growing population of cells. We calculated S-phase length for primary murine and human fibroblasts, *Drosophila* S2 cells, immortalized cells (NIH 3T3) and four human tumor cell lines (HCT116, HeLa, U2OS, Jurkat), demonstrating that this method is applicable to a variety of cell types of diverse origins. By doing so, we noticed that tumor cell lines commonly used by research laboratories tend to replicate their genome more slowly than non-tumor primary cells, as proposed in some old studies [42,43]. Of note, cancer cells also show higher variability in SPD measurements between experiments than untransformed cells (Figure 4b). This may be due either to their altered cell cycle control, truncated G_1_ phase and sub-optimal replication origin licensing [8,44], relaxed restriction point control, or to replication stress that slows down S-phase progression [45]. Further work will be necessary to determine if an extended S phase is a general property of tumor cells, and the causes of this extension. An advantage of our method is that it does detect the slowest S-phase progressing cells, and is theoretically amenable to heterogeneous populations that will appear as EdU^+^BrdU^+^/EdU^+^ regression lines of different slopes. This method of determining S-phase duration may then become a useful tool for tumor stratification and eventually to guide targeted therapies.

Our measurements of S-phase duration mainly agree with estimates from other studies, although a wide fluctuation exists in the literature, probably because indirect methods, cell synchronization or unwarranted assumptions have been applied in a number of such studies. The Fucci cell cycle reporter system, for example, estimates S-phase duration from the time between Cdt1 and Geminin degradation, not DNA synthesis directly [21]. PCNA-GFP, on the other hand, monitors the presence of foci to determine S-phase duration, but the precise time when DNA synthesis actually begins or ends is more difficult to assess using this reporter, as illustrated by the different SPD obtained for HeLa cells using PCNA-GFP videomicroscopy (7 h 30) or EdU-BrdU double labelling (11 h) [27]. Moreover, the possible deleterious effects of photo-damage during microscopy also have to be taken into account. Nonetheless the SPD we measured in *Drosophila* S2R+ cells is roughly the same as obtained by other groups (~300 min) using a mCherry-PCNA fusion [36] or the Fly-FUCCI system [37]. The shorter apparent S phase recorded for some S2 cells may reflect the fact that using hydroxyurea (HU) for synchronization does not block, but only slows down replication forks, with some cells entering G_2_ shortly after HU release [46]. The S-phase duration can be calculated from the S-phase cell fraction only if the cell cycle doubling time is measured accurately, which is not always easy due to heterogeneities in the cell population made of fast-cycling, slow cycling, paused, senescent and dead cells. The main advantage of the method described here is that it singles out the synthetically active fraction of cells, i.e., those able to incorporate EdU into DNA, and then follows these cells as they progress through and exit the S phase. It is therefore unaffected by the extended periods of time that different cells can be arrested by G_1_ or G_2_ checkpoints. It is believed that once cells enter S phase, they are committed to complete it, and the time it takes is precisely what is measured by our method. A similar method using dual EdU-BrdU pulsing and the Operetta cell analyzer has been published [47], but in this case the first EdU pulse was not terminated with a thymidine chase, and the fraction of cells exiting S phase (EdU^+^ BrdU^−^/EdU^+^ BrdU^+^) was measured at a single point (2 h). We believe that our method is more robust because cells entering S phase during the period between the two pulses are not labeled anymore with EdU (i.e the thymidine chase stops the increase in EdU^+^ cells), and because the fraction of cells exiting S is measured throughout the S phase instead of at a single time. For example, the fraction of cells entering and exiting the S phase (Figure 1c) is roughly equal during the course of the experiment shown in Figure 3, suggesting these cells transited regularly through S. The quality of the experiment and the regular S-phase progression of the cell population under study can be assessed by the good correlation coefficient to linear regression. Yet, we found that the SPD measured for a given cell type can vary and appear much longer in some experiments. In fact, we realized that this method is exquisitely sensitive to any physiological (confluence, senescence, poor quality serum), mutational (cancer cells) or chemical (drugs) inputs capable of causing an extended S phase, and is therefore ideally suited to detect the presence of replication stress in various cells and conditions. The cancer cell lines we analyzed showed consistently a higher coefficient of variation in S-phase duration between experiments compared to untransformed cells (Figure 4b), without evidence for heterogeneities within the cell population (good correlation in linear regression; Appendix A). This may stem from the relaxed G_1_ restriction point of cancer cells, allowing them to enter S phase in suboptimal conditions and variable kinetics.

Several precautions have to be taken when using this technique. First, as cell lines derive in culture, they might lose expression or function of thymidine kinase or nucleoside transporters, thereby becoming refractory to labeling with exogenous thymidine analogs. Second, the proliferation capacity and homogeneity of the culture are paramount for best results: since the method scores the slowest S-phase cells, a minority of slow cells can considerably extend the apparent SPD. Third, watching the evolution of EdU^+^ cells over time can be useful: if it increases during the chase, it means that more or fresher thymidine should be used for the chase, or SPD will be overestimated. Escape from nocodazole arrest will also cause SPD overestimation, but this can be detected by the BrdU^+^ fraction not reaching zero. On the other hand, a decreasing EdU^+^ fraction is indicative of cell loss (cell death or detachment of mitotic cells due to nocodazole arrest, especially in later time points). Collecting the rinsing buffer, possibly containing detached cells, can overcome this bias. Finally, some cells are resistant to nocodazole, for example *Drosophila* S2 and S3R+ cells; colchicine or other mitotic inhibitors can be used in this case.

In silico simulation of double-pulse staining for cell cycle analysis showed that the results should be more reliable than other existing methods, because of limited variance [48]. We confirm that our EdU-BrdU method allows robust measurements revealing reproducible differences between cell types, normal or cancer lines, although the latter showed increased variance. The rationale we designed is reminiscent of approaches used on flatworms [49] and rat fibroblasts [50], but because our method is based on flow cytometry rather than microscopy, we believe it is faster, more quantitative and amenable to many laboratories. The S-Phase fraction (SPF) in tumors has been proposed as a prognosis factor reflecting the uncontrolled proliferation of malignant cells. This measure gave conflicting results and its relevance to cancer prognosis was questioned [51]. Our study suggests that S-phase duration, rather than SPF, should be evaluated as a disease factor because it is indicative of replication anomalies, rather than merely the increase in the fraction of cycling cells. While more work is required to establish if a long SPD is a “hallmark of cancer”, and to discover what mechanisms or types of replication stress are responsible, we believe that the method described here should be of interest to a vast community of researchers studying cell cycle, development and diseases.

## Figures and Tables

**Figure 1 genes-13-00408-f001:**
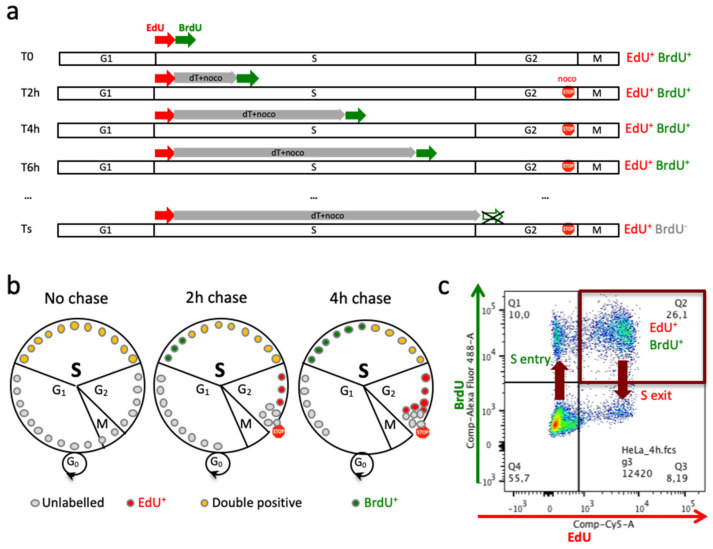
Principle of the method. (**a**) Asynchronous cells are pulse-labeled with EdU, then with BrdU for 30 min (T_0_). Other samples get the same two pulses, but separated by a thymidine chase period lasting 1.5, 3.5, 5.5 (T_2_, T_4_, T_6_); nocodazole is added to prevent progression to the next cycle. DNA synthesis time (T_S_) is the time when the earliest EdU^+^ cell is no more labelled with BrdU. (**b**) Illustration of cells (circles) progressing through the cell cycle, labelled with EdU (red), BrdU (green) or both (yellow), at T_0_ (no chase) or after intervening thymidine chases (2 h, 4 h). (**c**) An example of FACS analysis on HeLa cells double-labelled with 4 h thymidine chase. After staining with an anti-BrdU antibody and a fluorescent azide, double-positive, double-negative and single (EdU or BrdU) positive cells can be identified and scored by cytometry. EdU^−^ cells that become BrdU^+^ are those entering S phase during the interval between the two pulses (S entry); conversely, double-positive cells that become BrdU^−^ are those exiting S phase during the interval between the two pulses (S exit).

**Figure 2 genes-13-00408-f002:**
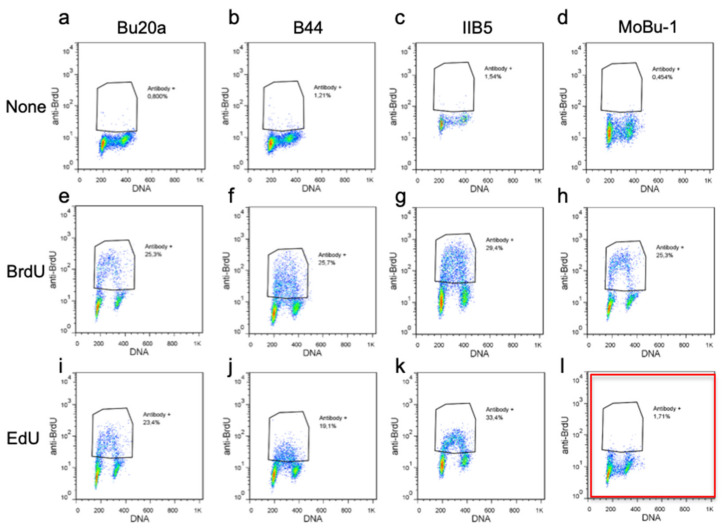
Selection of a BrdU antibody not recognizing EdU. Unlike the Bu20A (**a**,**e**,**i**), B44 (**b**,**f**,**j**) and IIB5 (**c**,**g**,**k**) anti-BrdU antibodies, clone MoBU-1 (**d**,**h**,**l**) does not cross-react with EdU-substituted DNA. MEF cells were left untreated (top row), treated with BrdU (middle row), or with EdU (bottom row), then stained with the various BrdU antibodies and detected by flow cytometry.

**Figure 3 genes-13-00408-f003:**
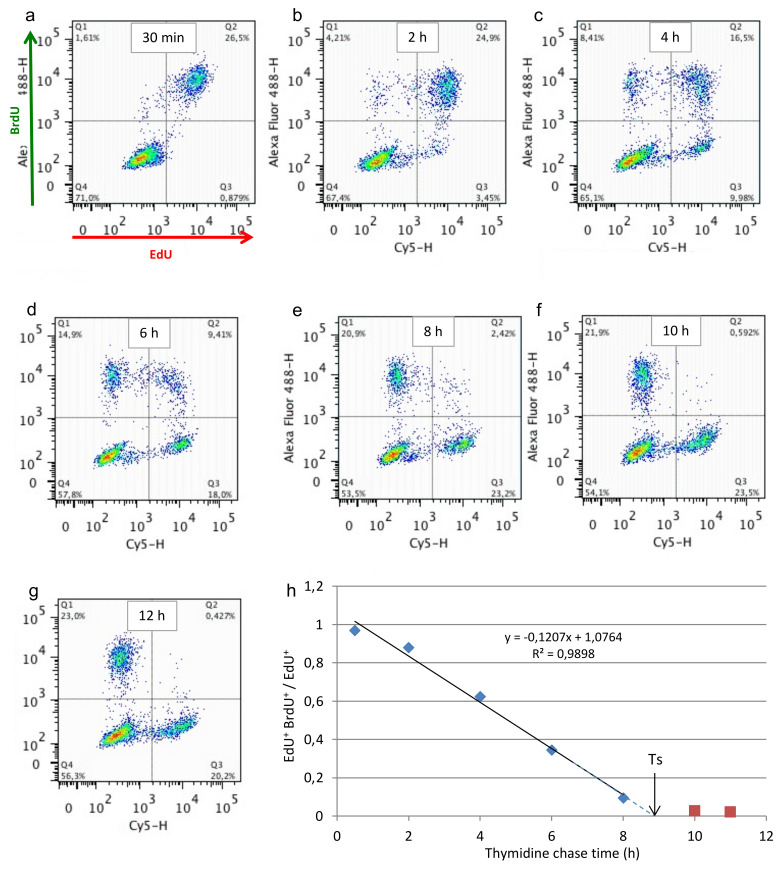
Asynchronous MEF cells were pulsed with 10 µM EdU, then with 100 µM BrdU (30 min each), either directly (**a**) or after a chase with thymidine and nocodazole of 1.5 h (**b**), 3.5 h (**c**), 5.5 h (**d**), 7.5 h (**e**), 9.5 h (**f**), and 11.5 h (**g**). Note double-positive cells (top right quadrant) decreasing over time. (**h**). Drawing a linear regression of the fraction of EdU^+^ BrdU^+^ cells amongst EdU^+^ cells over time determines DNA synthesis time (T_S_ or SPD) as the time when the regression line crosses the x axis.

**Figure 4 genes-13-00408-f004:**
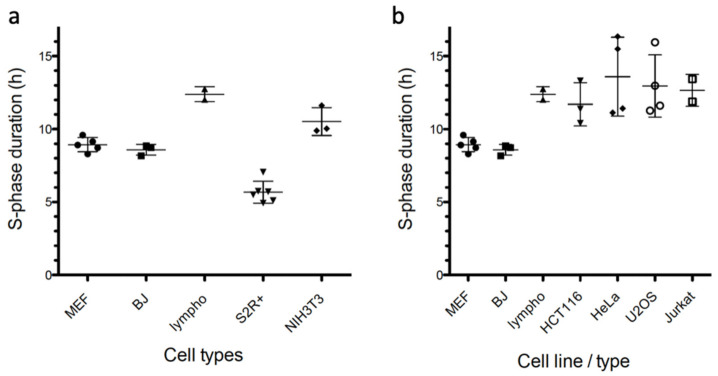
S-phase duration in different cell lines. (**a**) Proliferating cultures of mouse embryonic fibroblasts (MEF), human primary fibroblasts (BJ), human lymphocytes (lympho), Drosophila cell line (SR2R+) and immortalized mouse fibroblasts (NIH3T3) were treated as indicated and SPD measured. Each dot represents an independent biological replicate. (**b**) S-phase durations obtained for primary (MEFs, BJs, lymphocytes) and cancer-derived (HCT116, HeLa, U2OS, Jurkat) cells.

**Figure 5 genes-13-00408-f005:**
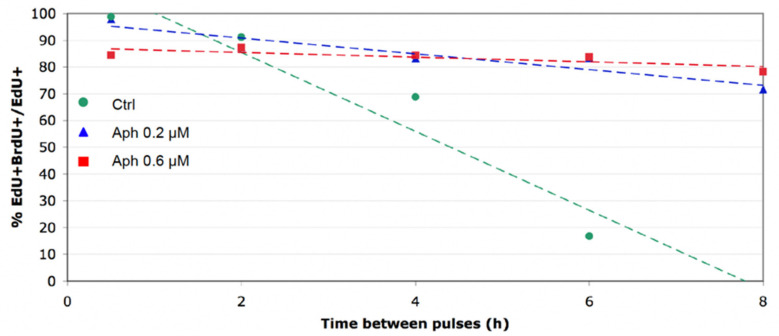
Effect of aphidicolin on S-phase duration. Linear regression of the fraction of EdU^+^ BrdU^+^/EdU^+^ cells after treating MEFs with no (Ctrl), 0.2 µM or 0.6 µM aphidicolin. Estimated S-phase durations are 8.5 h, 31 h and 124 h, respectively.

**Table 1 genes-13-00408-t001:** Measured duration of S phase in normal and cancer cell lines.

	MEF	BJ	Lympho	HCT116	HeLa	U2OS	Jurkat
Mean (h)	8.94	8.59	12.40	11.70	13.60	12.95	12.66
*n*	5	3	2	3	4	4	2
SEM	0.22	0.22	0.36	0.85	1.35	1.06	0.77

## Data Availability

All primary data are available upon request.

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
