# Peer review of "Measuring S-Phase Duration from Asynchronous Cells Using Dual EdU-BrdU Pulse-Chase Labeling Flow Cytometry"

_genes, 2022, doi:10.3390/genes13030408_

Round 1
Reviewer 1 Report
The manuscript by Bialic et al describes a method based on dual EdU-BrdU labeling to measure the duration of S-phase by flow cytometry. Although related approaches are used to analyze cells in different cell cycle phase, the implementation of a time-course chase between both labeling allows a reliable measurement of S-phase duration. This method has already been successfully used in the context of a collaboration with the A.Castro/T. Lorca laboratory. The technique is relatively simple and may be applied in any standard laboratory, as it does not require expensive compounds or fancy equipment. Its strength is certainly that no cell synchronization is required and that it provides a precise measurement of S-phase duration.
The authors present measurements of different types of cells and found that cancer-derived cell lines show a longer S-phase than primary cells. It appears particularly striking to me that those cancer-derived cell lines also show a much higher variability between experiments. It would thus be interesting to provide a more detailed description of these experiments to assess whether the variability is already detectable within a cell population (linear regression of the measurement points, outliers) or only between experiments.
Overall, the manuscript presents a novel and useful method to measure S-phase duration. Moreover, the method gives insights into possible S-phase perturbations, as exemplified with cancer-derived cell lines.
Minor points:
Figure 5: The resolution of the figure is poor and the fist point of the control culture is outside of the graph (above 100%?).
Page 5, last paragraph: The authors determined the minimal required amounts of EdU and BrdU for a good detection. It might be useful for the readers to know whether these amounts work well for all the cell types that have been used or whether the amount of analogues needs to be adjusted depending on the type of culture.
Page 1, lanes 31-34: prepositions (during/in) are missing between ‘activity’ and ‘late M…’ and ‘S phase…’
Page 8, lane 277: Should read: ‘….our finding that several cancer----‘
Author Response
We thank reviewer 1 for his/her support and insightful comments or suggestions. We have modified the manuscript as follows:
- SPD variability in cancer cell lines: the larger variability of SPDs measured between experiments in cancer cell lines could indeed stem from a prexisting heterogeneity within the population. If this were the case, one could expect irregularities in the linear regression of EdU+BrdU+ cells over time, or changes in the slope reflecting the co-existence of fast- and slow-replicating cells in the population. We compared all FACS data and and provide below linear regressions of EdU+ BrdU+ cells. We found no significant difference between correlation coefficients (R2) for normal and transformed cell lines (see page 3 of SuppFig1). This suggests that cancer cells progress uniformly through S phase within a population, but show a higher variability in measured SPDs between experiments. We don't know the reason for this, which may be due to a higher sensitivity of cancer cells to internal (cell population history) or external cues, related to their relaxed restriction point control and S-phase entry under non-optimal conditions. This peculiarity is now commented in the discussion (line 387). We wish not to expand more on that issue in this methods article, as a more complete description of replication dynamics in cancer cells will be published elsewhere.
- Figure 5: its resolution has been improved. The first point for Ctrl is just below 100%, not above. However, the linear fit of EdU+ BrdU+ cells for Ctrl extrapolates above 100% at time zero, which is due to suboptimal analog detection in this experiment. We believe it does not change the message of this figure, which is that drug-induced S-phase slowing is measurable using this method.
- It is now stated (line 257) that the same concentrations of EdU, thymidine and BrdU were used for all the cell lines tested.
- Typographical errors have been corrected (lines 32 and 282). Thank you.

Reviewer 2 Report
In the manuscript, the authors have described a method for measuring the
length of the S phase. The length of the S phase is particularly of interest
because it has a strong connection with cancer. It is well known that
oncogenic activation reduces the length of G1 by inducing premature S phase
entry, which in turn not only induces conflicts between transcription and
replication leading to genomic instability, a hallmark of cancer but also
increases the length of the S phase. As a result, the length of the S phase
potentially could be a reasonable measure of oncogenic transformation.
That is why measuring length of the S phase has been of great interest in the
cancer field and several methodologies have been developed. Having said
that, the current manuscript described a novel approach of measuring the S
phase by dual labeling of DNA synthesis, which doesn't require any
synchronization. This method will be of high interest for scientists working not
only in cell biology but also in the cancer biology field. The manuscript is well
written, well structured and could very well be published with some minor
improvements.
In general, several places need a reference. The authors should be
acknowledging previously published work in this area, given several studies
have previously measured the length of the S phase. It would be great for the
readers if the methods were a bit more descriptively written.
Other than that please find some minor suggestions bellow, which the authors
might consider including.
Line 9
Abstract: Eukaryotes duplicate their chromosomes during the S phase using
Line 13
Methods to measure the S phase duration were so far indirect, based on
Line 18
engineering, thus avoiding possible artefacts. It measures the duration of
Line 27
proliferation and tissue development. Cells have evolved several mechanisms
to
Line 40
targeted by oncogenic mutations, resulting in processes collectively referred
to
Line 71
difficult and potentially leads to an overestimation of the SPD. Finally, the
direct observation of cells progressing through the S phase by videomicroscopy
of
Line 74
[21], or the use of cell-permeant replication tracer [22]. In both cases, exposure
Line 84
provides the duration of the S phase. We believe that this method will be
useful
Line 115
EdU (Carbosynth) for 30 min, the medium removed and replaced with a new
Line 131
A FACSCanto II (Becton-Dickinson) with 405, 488 and 633 nm laser lines
were
Line 150
With this in mind, we thought to first label briefly the fraction of cells in the S
phase with EdU, chase it with thymidine for varying times, and finally, label
the cells that are still in the S phase with BrdU.
Line 156
labeled with the second analog corresponds to the duration of the S phase
(Ts). Nocodazole, which prevents spindle formation, is added along with
thymidine to prevent cells from entering mitosis and getting labeled with BrdU
in the subse- quent S phase. Provided that each analog can be detected
specifically, it would be straight- forward to quantitate by flow cytometry the
fraction of double-positive cells, their de- crease with longer chase times, as
well as the flux of cells entering and exiting the S phase (Figure 1c).
Line 183
This in agreement with earlier studies [24,25] prompted us to use the MoBU-1
antibody for further experiments.
Line 198
sufficient for a clear distinction between EdU+ and EdU- cells by cytometry,
with
Line 235
that the S phase was completed and that cells did not enter the S phase of
the subsequent cell cycle. The ability to follow the population of S-phase cells
as they progress through and exit S over time allows determining the duration
of the S phase with good precision. Moreover, the good linear regression
fitting indicates that S-phase progression was uniform in the cell population.
The method is robust; the mean (±SEM) S-phase duration was found to be
8.94 h (± 0.22, n=5) for MEF.
Line 265
cycle and may affect the dynamics of the S phase. Replication stress has also
been
Line 268
normal or deregulated S phase duration. Interestingly, the S phase lasted
Line 273
SPD (12.66 h ±0.77, n=2), but in this case, it was similar (12.4 h ±0.36, n=2)
to Line 277
Nonetheless, our finding that several cancer cell lines display a prolonged S
Line 283
robust, we next considered using it to detect and quantify the extent of
Line 288
at which EdU+ MEF cells exited the S phase, compared to un- treated control
Line 294
responses [32]. Our estimation of the S-phase duration in aphidicolin-treated
Line 304
flow cytometry, to measure the duration of the S phase in an asynchronously
Line 315
progressing cells, and is theoreti- cally amenable to heterogeneous
populations Line 325
applied in a number of such studies. The Fucci cell cycle reporter system, for
Line 328
PCNA-GFP, on the other hand, monitors the presence of foci to determine SLine
341
and then follow these cells as they progress through and exit the S phase. It is
therefore unaffected by the extended periods of time that different cells can be
arrested by G1 or G2 checkpoints. It is believed that once cells enter the S
phase,
Line 346
Operetta cell analyzer has been published [36], but in this case, the first EdU
Line 348
believe that our method is more robust because cells entering the S phase
during the period between the two pulses are not labeled anymore with EdU
(i.e the thymidine chase stops the increase in EdU+ cells), and because the
fraction of cells exiting the S is measured throughout S phase instead of at a
single time. For example, the fraction of cells entering and exiting the S phase
(Figure 1c) is roughly equal during the course of the experiment shown in
Figure 3, suggesting these cells transited regularly through the S phase. The
Author Response
We thank reviewer #2 for her/his laudatory comments on the usefulness of a robust method to measure the length of S phase during the process of oncogenic transformation.
For the comments:
- We now added several references of papers where the length of S phase has been measured using similar or different methods. As stated by all reviewers, the advantage of this method is that it does not require cell synchronization, gene editing and measures nucleotide incorporation directly.
- We now provide in Annex 1 a step-by-step protocol for future users, which contains all details and should make reproduction easy.
- The vast majority of reviewer's 2 grammatical corrections have been approved and changed in the text. Thank you for this.
Reviewer 3 Report
This work describes a simple method to quantify S-phase duration amenable to use in most cell culture types, which is of potential interest for the molecular and cell biology community. The manuscript is concisely written, the data are clear and well presented, and the advantages/disadvantages of the method in comparison with the existing literature are accurate. Some points of improvement are:
- The layout of Figure 5 can be improved. On one hand, the quality of the plot is very low. On the other, it would be of interest to show longer time points of the Aph-treated cells; is it possible to detect S-phase exit (EdU+ only cells) with this Aph doses? Also, why the linear regression fitting for the untreated cells is so poor in this experiment in comparison with the one in Figure 3?
- English native correction of the summary will be beneficial to engage readers
Author Response
We thank reviewer 3 for his/her comments, which were taken into account.
Specifically,
- We uploaded a higher resolution Figure 5
- Unfortunately, we did not extend the time points of Aphidicolin-treated samples enough to cover entirely this much slowed down S phase. However, at 0.2µM Aph, we see about 20% of the cells exiting S phase after 8h, indicating that DNA replication is not arrested in these cells. This is consistent with data from Koundrioukoff et al (2013) who found that fork speed is reduced ~5-fold without activating a checkpoint response, in this condition.
- We agree and apologize for the sub-optimal linear regression for Ctl cells in this experiment (Fig.5), the reason of which is unknown to us. However points align rather well for the Aph 0.2 and 0.6 µM experiments, which we feel illustrates adequately that this method is able to detect S-phase extension due to fork slowing.
- We tried to improve the english in the summary, as in the text, according to reviewers suggestions.